# A Phenomenological Model for Enthalpy Recovery in Polystyrene Using Dynamic Mechanical Spectra

**DOI:** 10.3390/polym15173590

**Published:** 2023-08-29

**Authors:** Koh-hei Nitta, Kota Ito, Asae Ito

**Affiliations:** Division of Material Sciences, Graduate School of Natural Science and Technology, Kanazawa University, Kakuma Campus, Kanazawa 920-1192, Japanasae@se.kanazawa-u.ac.jp (A.I.)

**Keywords:** enthalpy relaxation, dynamic mechanical spectra, creep compliance, polystyrene

## Abstract

This paper studies the effects of annealing time on the specific heat enthalpy of polystyrene above the glass transition temperature. We extend the Tool–Narayanaswamy–Moynihan (TNM) model to describe the endothermic overshoot peaks through the dynamic mechanical spectra. In this work, we accept the viewpoint that the enthalpy recovery behavior of glassy polystyrene (PS) has a common structural relaxation mode with linear viscoelastic behavior. As a consequence, the retardation spectrum evaluated from the dynamic mechanical spectra around the primary *T_g_* peak is used as the recovery function of the endothermic overshoot of specific heat. In addition, the sub-*T_g_* shoulder peak around the *T_g_* peak is found to be related to the structural relaxation estimated from light scattering measurements. The enthalpy recovery of annealed PS is quantitatively described using retardation spectra of the primary *T_g_*, as well as the kinetic process of the sub-*T_g_* relaxation process.

## 1. Introduction

When an amorphous polymer is cooled from well above glass transition temperature *T_g_* to some lower temperature, rotational mobility around the main-chain bonds is frozen, and the polymer has no time to attain conformational equilibrium within the time scale of the given experiment. As a result, the glassy polymer solidified below *T_g_* usually shows time-dependent physical properties such as heat capacity *C_p_*, modulus, and density. This is a feature known as “physical aging” [1,2,3]. Then, a series of atomic rearrangements of the main chain towards the new equilibrium state proceeds in order to lose the excess configurational energy, and the subsequent time-dependent change is referred to as “structural relaxation” [4,5,6,7].

Figure 1 illustrates schematical plots of enthalpy *H* and heat capacity *C_p_* of a glassy polymer under cooling and subsequent reheating. The glass transition temperature *T_g_* has been conventionally defined as a temperature that is the intersection of the slopes of the liquid state and the glassy one. The maintenance at temperature *T_a_* below *T_g_* after cooling from a liquid state induces enthalpy relaxation toward the equilibrium value. This phenomenon is called enthalpy relaxation or physical aging. The equilibrium state is represented by the extension of the slope of the liquid state (the dashed line in Figure 1). A cross point of the dashed line and the *H*–*T* curve gives a limiting fictive temperature $T_{f}^{0}$. The enthalpy *H* overshoots and deviates downward from the equilibrium line during heating after enthalpy relaxation. The enthalpy recovery behavior can be detected as a peak around the glass transition during reheating on the *C_p_*–*T* curve.

The experimental technique most frequently used for the characterization of structural relaxation is the measurement of enthalpy changes by differential scanning calorimetry (DSC). The internal rotation of the main chain toward reducing configurational energy, i.e., structural relaxation, emerges as an overshoot in the *C_p_* curve under a constant heating rate after annealing at a temperature below *T_g_*. The overshoot peak area resulting from the difference in both corresponding *C_p_*–temperature lines is in accordance with the enthalpy difference $∆H_{\mathrm{DSC}}$ between the extrapolated enthalpy of the liquid and that of the glass. The overshoot in the *C_p_* curve shows a positive dependence on the annealing temperature and time. This thermodynamic feature is known as “enthalpy relaxation” [8,9,10]. Consequently, the changes in the enthalpy difference ∆H with annealing time at a fixed annealing temperature can be expressed using a decay function $\phi \left(t\right)$ of the annealing time *t* as follows:(1)$$∆H_{\mathrm{DSC}}=∆H_{e}\left(1-\phi \left(t\right)\right)$$
and here we introduce the normalized recovery function $\Lambda_{T}\left(t\right)$ as:(2)$$\Lambda_{T}\left(t\right)=1-\phi \left(t\right)$$

The enthalpy loss $∆H_{DSC}=H_{0}-H\left(t\right)$, where $H_{0}$ is the initial enthalpy at the annealing time t=0, and $H\left(t\right)$ is the enthalpy at any annealing time *t*, and the final loss $∆H_{e}=H_{0}-H_{\infty }$, where $H_{\infty }$ is the enthalpy at equilibrium (t=∞), and $\phi \left(t\right)$ is an exponential decay function. It has been identified that the decay function $\phi \left(t\right)$, describing the kinetics of enthalpy recovery, is characterized by two essential features: non-exponentiality and non-linearity. The non-exponential character is brought about by the distribution of relaxation times, which can be usually described by the Kohlaush–Williams–Watts (KWW) equation [11,12,13] as follows:(3)$$\phi \left(t\right)=e^{-{\left(t/\tau_{0}\right)}^{\beta }}$$
where $\beta \left(0<\beta <1\right)$ is a parameter associated with the distribution of characteristic time $\tau_{0}$. The small value of β implies a broad distribution, and β=1 implies a single characteristic time. Additionally, the non-linearity is caused by the discrepancy of the approach from below and from above an equilibrium point. This discrepancy is revealed by the double dependence of $\tau_{0}$ on both the actual temperature *T* and the fictive temperature *T_f_* in Equation (3) [14,15], which is defined as the temperature at which the glass would be in the equilibrium structural state if it was instantaneously brought to the temperature. In this view, a non-linearity parameter $x\left(0<x\leq 1\right)$ is introduced to partition the activation energy into the actual temperature and the fictive temperature [16,17] as follows:(4)$$\tau_{0}=A\exp\left[x∆h/RT+\left(1-x\right)∆h/RT_{f}\right]$$
where *A* is an arbitrary relaxation time, ∆h is activation energy, and *R* is the gas constant. 

So far, much effort has been made to develop mathematical models that permit the comparison of experimental *C_p_* data obtained after various thermal histories with theoretical predictions. Some fitting parameters in these phenomenological models are used to predict the overshooting behavior of the experimental *C_p_* curve during a constant rate of heating [16,17,18,19,20,21,22,23,24,25]. The Tool–Narayanaswamy–Moynihan (TNM) model [15,26,27] has been most widely used to describe the enthalpy relaxation behavior. Here, we outline the enthalpy relaxation in the TMN model under a constant heating rate test.

A DSC experiment under continuous heating/cooling can be treated as a series of infinitesimal temperature jumps (*T*-jumps). In general, then, the response of the fictive temperature to the departure of the enthalpy from its equilibrium value follows the recovery function of the enthalpy loss:(5)$$T_{f}=T_{0}+∆T\Lambda_{T}\left(t\right)$$
where *T*_0_ is the initial temperature at t=0, and ∆T is the magnitude of the *T*-jump. Taking the relaxation function to be linear with respect to *T*-jumps, the net response of the system can be formulated as the superposition of the response to the series of *T*-jumps that constitute their thermal histories. Combination of Equations (1) and (3) with the Boltzmann superposition principle leads to the evolution of fictive temperature as a function of the actual temperature:(6)$$T_{f}\left(T\right)=T_{0}+\int_{0}^{t}dt^{\prime }{\left(\frac{dT}{dt}\right)}_{t=t^{\prime }}\left(1-\exp\left[-{\left(\int_{t^{\prime }}^{t}dt^{\prime \prime }\frac{1}{\tau_{0}}\right)}^{\beta }\right]\right)$$
where *t*′ and *t*″ are dummy integral variables of time. Differentiation of Equation (6) yields the normalized specific heat capacity as:(7)$$C_{p}^{N}\left(T\right)=\frac{dT_{f}}{dT}$$

This mathematical process has been used to give a description of the specific heat as a function of actual temperature. TNM-based models simulate the enthalpy relaxation in a series of various polymeric materials by controlling fitting parameters such as *A*, *x*, ∆h, and β. 

The recovery function $\Lambda_{T}\left(t\right)$, describing the enthalpy changes, is a retardation function rather than a relaxation one. Therefore, the characteristic time $\tau_{0}$ in Equation (3) should be referred to as “retardation time” rather than “relaxation time”, and, in many endothermic experiments described as “enthalpy relaxation”, the “enthalpy recovery” is recorded as a function of temperature or time, as suggested by Hodge [8]. This implies that it is not correct to directly compare DSC and dynamic mechanical analysis (DMA) [28].

In this work, we propose a novel phenomenological model equation as a natural extension of the linear viscoelasticity to simulate the overshoot of *C_p_* under constant heating rate as a function of annealing time *t* at an annealing temperature below sub-*T*_g_ for atactic polystyrene. Struik also pointed out the experimental fact that physical aging occurs in the temperature range of *T*_g_ to sub-*T*_g_ [1]. The retardation time function can be straightforwardly applied to the enthalpy recovery to give the enthalpy *H* or specific heat *C_p_* curves as a function of the annealing time. The present work is fundamentally based on the concept that the time-dependent behavior of enthalpy recovery and dynamic mechanical spectra is associated with a common structural relaxation mode in the glassy state.

The relationship between the time scale of enthalpy relaxation and that of some accompanying mechanical responses such as creep, stress–relaxation, and stress–strain behavior has been widely studied by a number of workers [29,30,31,32,33,34,35,36]. However, there does not seem to have been any universal behavior found between thermo and mechanical dynamics, as suggested by Simon et al. [31] and Robertson et al. [33]. This is because much effort has been made to interrelate both time-dependent behaviors via the decay function $\phi \left(t\right)$. Yoshida [30] demonstrated that the enthalpy recovery rate after annealing corresponds to the change in dynamic storage modulus with annealing for amorphous and semicrystalline engineering plastics. A series of configurational rearrangements of the main chain reflecting the strain evolution towards the new equilibrium state under a creep test in the linear viscoelastic region corresponds to the structural rearrangement reflecting the enthalpy recovery during heating after a fixed annealing process to lose the excess configurational energy. In this work, on the basis of this view, we expand the TNM model to reproduce the *C_p_* overshooting peak due to the enthalpy recovery process as determined from DSC experiments where we used retardation spectra and kinetic data of relaxation peaks from linear dynamic mechanical data in place of some fitting parameters of the TNM model. 

This paper is organized as follows. In Section 2, we propose a phenomenological theory for expressing the evolution of fictive temperature in which the retardation time spectrum of viscoelastic properties is incorporated in the recovery response in the TNM model. Section 3 shows the experimental procedure for monitoring the annealing in PS by DSC and DMA. The enthalpy recovery data and DMA spectra are summarized in Section 4. Section 5 compares the experimental DSC curves for the enthalpy recovery behavior of various annealed PSs with the curves predicted from their DMA spectra. We give concluding remarks in Section 6.

## 2. Theory

In the case of the TNM model, the Boltzmann superposition principle is applied to determine the response to thermal histories. Consider first the fictive temperature *T_f_* response to a sequence of *T*-jumps $∆T_{j}$ applied at time $t_{j}$. Then, the $T_{f}$ response at any time *t* becomes the sum of the recovery response $\Lambda_{T}\left(t\right)$ given by Equation (2) for each *T*-jump:(8)$$T_{f}\left(t\right)=T_{0}+∆T_{1}\Lambda_{T}\left(t-t_{1}\right)+∆T_{2}\Lambda_{T}\left(t-t_{2}\right)+∆T_{3}\Lambda_{T}\left(t-t_{3}\right)+\cdots$$

The Riemann integral form of Equation (8) corresponds to Equation (6).

An analogous derivation can be made for the response of strain to a sequence of stress steps on the basis of a corresponding principle; thus, each stress step makes an independent contribution to the final strain in which incremental stresses $∆\sigma_{1},∆\sigma_{2},∆\sigma_{3,}\;etc.,$ are added at times *t*_1_, *t*_2_, *t*_3_, etc., respectively. The total strain $\varepsilon \left(t\right)$ is obtained by the addition of all the contributions:(9)$$\varepsilon \left(t\right)=∆\sigma_{1}D\left(t-t_{1}\right)+∆\sigma_{2}D\left(t-t_{2}\right)+∆\sigma_{3}D\left(t-t_{3}\right)+\cdots$$
where $D\left(t\right)=\varepsilon (t)/\sigma$ is the tensile compliance as a function of time. It should be noted here that the compliance $D\left(t\right)$ is independent of applied stress. As a result, the formulation of the Boltzmann superposition principle for a multistep creep of linear viscoelastic solids generalizes the summation of Equation (9) to be the Riemann integral form.
(10)$$\varepsilon \left(t\right)=\int_{-\infty }^{t}dt^{\prime }{\left(\frac{d\sigma }{dt}\right)}_{t=t^{\prime }}D\left(t-t^{\prime }\right)$$

Here, we assume that a stress history initiates at t=0. In the general case where there exists a distribution in retardation time τ, the compliance is then given by:(11)$$D\left(t-t^{\prime }\right)=\int_{-\infty }^{\infty }d\ln\tau L\left(\ln\tau \right)\left\{1-\exp\left[-\left(t-t^{\prime }\right)/\tau^{\prime }\right]\right\}$$
where $\tau^{\prime }$ is the retardation time at *t′*, and $L\left(\ln\tau \right)$ is the retardation time spectrum. Here, we assume that the enthalpy recovery behavior has a common structural relaxation mode with linear viscoelastic behavior. It follows that the recovery function $\Lambda_{T}\left(t\right)$ in enthalpy corresponds to the retardation function in creep strain. Consequently, we have a phenomenological equation for expressing the evolution of fictive temperature as the Riemann integral form:(12)$$T_{f}\left(t\right)=T_{0}+\int_{0}^{t}dt^{\prime }{\left(\frac{dT}{dt}\right)}_{t=t^{\prime }}{\int_{-\infty }^{\infty }dln\tau L^{N}}^{}\left(ln\tau \right)\left\{1-\exp\left[-\left(t-t^{\prime }\right)/\tau^{\prime }\right]\right\}$$
where $L^{N}\left(\ln\tau \right)$ is a normalized retardation spectrum defined by:$$L^{N}\left(\ln\tau \right)=L\left(ln\tau \right)/\int_{-\infty }^{\infty }dln\tau L\left(ln\tau \right)$$

Thus, the enthalpy recovery curves can be obtained only from the viscoelastic data without any fitting parameters.

## 3. Experimental

### 3.1. Sample Preparation

Pellets of atactic polystyrene (PS) with a molecular weight *M_w_* = 28 × 10^4^ and *M_n_* = 11 × 10^4^ were comp-molded at 230 °C under 40 MPa to films about 0.2 mm thick for DSC and DMA experiments. Then, the comp-molded samples were cooled to 25 °C at around −40 °C/min. The cooled specimens were annealed for 1, 5, 10, and 24 h in a vacuum oven at 80 °C for DMA measurements.

### 3.2. DSC Measurements

The DSC measurements were performed using a Perkin Elmer Diamond DSC with a controlled cooling accessory (Intracooler 2P) under nitrogen gas flux (flow rate of about 4.5 psi and 20.0 mL/min). The temperature and heat capacity data were calibrated with standard indium and sapphire, respectively. The samples of about 3 mg, cut out from the slow-cooled films, were sealed in aluminum pans. Figure 2 shows the thermal history for the measurements of the dependence of enthalpy relaxation on the annealing time. The sample was kept at 230 °C for 10 min and cooled to an annealing temperature of 80 °C ($=T_{g}-20\;$°C) at −40 °C/min, kept for annealing time of 1, 5, 10, and 24 h, further cooled to 50 °C, and then heated to 230 °C at 20 °C/min for monitoring of enthalpy recovery.

### 3.3. DMA Measurements

The temperature dependence of storage and loss moduli of the film specimens were examined in the temperature range from 50 °C to 150 °C at a heating rate of 2 °C/min under fixed frequencies of 10, 30, 100, and 200 Hz under a nitrogen gas atmosphere using a dynamic mechanical analyzer (UBM DVE-V4). The rectangular sample specimens with 3 mm width, 0.2 mm thickness, and 20 mm length were cut out from the slow-cooled and annealed films. The grip-to-grip distance was 10 mm, and the dynamic amplitude was set to be 1 μm for the dynamic mechanical analysis (DMA) measurements.

In addition, the dynamic mechanical data over a wide range of frequencies were measured. The dynamic mechanical spectrum in the glass transition region was measured by the elongation mode in the temperature range from 70 °C to 110 °C and the frequency range from 0.5 to 300 Hz using DVE-V4 using the rectangular sample specimens with 3 mm width, 0.2 mm thickness, and 20 mm length. The viscoelastic properties from the glass transition region to the flow region were measured using the strain-controlled rotary rheometer; the measurements were made using a strain-controlled rotational rheometer (UBM G300NT) with a parallel plate of 25 mm diameter and a gap distance of 0.5 mm. A temperature range of 110 to 230 °C, a frequency range of 0.01 to 20 Hz, and a displacement of 0.5° were applied for the measurements. It should be noted that the displacement was set as 0.1° because the sample showed a glassy state. The shear modulus *G* values obtained directly by the rheometer were converted to the tensile modulus *E* values using E=3G.

### 3.4. Light Scattering Measurements

To quantitatively examine the effects of annealing on heterogeneous molecular aggregation, we carried out small-angle light scattering (SALS) measurement with a light scattering photometer (IST Planning SALS-100S) using a He–Ne laser with stabilized power supply (NF EC1000S). The detector used was a photomultiplier (H10721-20) that can rotate horizontally around a film specimen to scan scattering angles from 30° to 60°. The transmitted intensity was amplified by IST photosensor and recorded by a transient computer memory every 0.2° as a function of angle. 

The well-collimated incident beam was polarized in any direction in a plane perpendicular to the incident beam by using a polarization rotator and monochromatized to have a wavelength of 632.8 ± 2.4 nm using a band-pass filter. An analyzer was placed between the film specimen and the photomultiplier whose polarization direction was also rotatable in a plane perpendicular to the incident beam, thus, achieving any combination of polarized scatterings, such as *V_V_* (vertical polarizer, vertical analyzer) and *H_V_* (vertical polarizer, horizontal analyzer) polarizations.

## 4. Results

The set of DSC second run curves after an isothermal annealing stage at 80 °C was examined. All of the curves measured after annealing showed an overshoot peak at a temperature higher than *T_g_*. The main feature of these experiments was that the area and the position of the overshoot increased with increasing annealing time. From the original DSC second run curves, it was possible to evaluate the normalized specific heat $C_{p}^{N}(T)$ from the specific enthalpy lines corresponding to the liquid and glassy states as:(13)$$C_{p}^{N}\left(T\right)=\frac{C_{p}-C_{p}^{glass}}{C_{p}^{liquid}-C_{p}^{glass}}$$

The limiting fictive temperature $T_{f}^{0}$ [27] was determined as the extrapolated intersection of the pre-transition (or glass region) and post-transition (or liquid region) in enthalpy units, as indicated by point (f) of Figure 1. The method for determining $T_{f}^{0}$ of PS annealed for 24 h at 80 °C is exemplified in Appendix A. The fictive temperature $T_{f}$, as shown in Figure 3, was obtained by adding $T_{f}^{0}$ to the integration $C_{p}^{N}(T)$ with time (or temperature) using Equation (7). In addition, the normalized enthalpy loss $∆H^{N}\;$ could be calculated from $C_{p}^{N}(T)$ data as a function of the annealing time. It is confirmed in Figure 4 that the enthalpy recovery proceeded toward an equilibrium value as the annealing time increased.

Temperature dependences of dynamic storage modulus *E*′ and loss modulus *E*″ measured at 10 Hz are shown in Figure 5. We observed a shoulder peak (sub-*T_g_*) at some lower temperatures of primary peak (*T_g_*). It should be noted here that the primary *T_g_* relaxation peak was almost insensitive to the annealing time, whereas the magnitude of the sub-*T_g_* shoulder changed with the annealing time. The *E*″ values for other frequencies plotted against the inverse of temperature are exemplified for PS annealed for 1 h in Figure 6.

The appearance of similar sub-*T_g_* peaks in various polymers on the endothermic curves below *T_g_* was reported in several articles [16,37,38,39,40]. At longer annealing time or aging, sub-*T_g_* endotherm peaks approach limiting values as the annealed or aged glass approaches the equilibrium state, and the sub-*T_g_* peaks evolve into overshoots [8]. In addition, such sub-*T_g_* endotherms and mechanical spectra have been observed in glassy polymers subjected to varied pressure and mechanical stress [41,42,43,44]. The secondary or subglass relaxation has been considered to reflect the transformation of the non-thermal perturbation such as the non-homogeneous (non-equilibrium) structure resulting from quenching from *T_g_* into the homogeneous (equilibrium) structure. Wypych et al. [45] found from low-frequency Raman scattering data of a glassy PMMA that the aging makes the nano-scale structural aggregation more homogeneous, leading to an equilibrium structure with lower energy. 

To characterize the position, the shape, and the relaxation strength of sub-*T_g_*, we analyzed the *E*″ peak in the glass transition region by peak decomposition using the Cole–Cole function given by [46]:(14)$$E^{\prime \prime }\left(\omega ,\tau \right)=\left(E_{U}-E_{R}\right)\frac{{\left(\omega \tau \right)}^{\beta }\sin\left(\beta \pi /2\right)}{1+2{\left(\omega \tau \right)}^{\beta }\cos\left(\beta \pi /2\right)+{\left(\omega \tau \right)}^{2\beta }}$$
where *E_U_* and *E_R_* represent the limiting values of the storage modulus at infinite and zero frequency, respectively. In this function, the broadening of *E*″ is reflected by a decrease in β to values lower than unity. One example of the fitting to the dynamic glass transition is given in Figure 6. All other fitting data are shown in Appendix A of the supporting information.

The decomposition was performed on the dynamic loss modulus as a function of *T*^−1^ and possibly into one primary *T_g_* and two individual peaks, sub-*T_g_*_1_ and sub-*T_g_*_2_, appearing in the range from 90 °C to 100 °C. The frequency dependences of the primary peak and two sub-*T_g_* peaks give their activation enthalpy values under the Arrhenius activation law. All the data complied with the Arrhenius relation and gave rise to the apparent activation enthalpies shown in Figure 7. It is interesting to note that the activation enthalpies of both sub-*T_g_*s exponentially decayed with increasing annealing time, whereas that of primary *T_g_* was almost constant (about 500 kJ/mol). The Arrhenius plots are shown in Appendix A.

The master curve of frequency dependences with the reference temperature (*T*_r_) of 100 °C is exemplified for pristine PS in Figure 8. The thermos-rheological simplicity was confirmed to be established for the PS sample for a wide range of frequency. The starting numeral of *E*′ and *E*″ denotes the measurement temperature in °C.

*V_V_* and *H_V_* light scattering intensities are plotted against the scattering angle *θ* for annealed samples in Figure 9. The *H_V_* scattering intensities are almost independent of angle *θ*, but the *V_V_* scattering intensity negatively depends on *θ*, and its magnitude decreases with increasing annealing time, suggesting the existence of a non-thermal ordered structure in the glassy PS matrix. In addition, the size is postulated to be enough to cause observable angle dependence over the accessible range of the light scattering vector $\left|s\right|=2\sin\left(\theta /2\right)$. The *V_V_* intensity composed of the angle-dependent “excess” $V_{V}^{ex}(\theta )$ term superimposed on angle-independent “background” $V_{V}^{0}$ and *H_V_* scattering terms is as follows [47]:(15)$$V_{V}=V_{V}^{ex}(\theta )+V_{V}^{0}+\frac{3}{4}H_{V}$$

The angle-dependent $V_{V}^{ex}(\theta )$ resulting from thermally induced density fluctuations is given by [48]:(16)$$V_{V}^{ex}(\theta )=\frac{\pi^{2}}{9n^{4}\lambda^{4}}{\left(n^{2}-1\right)}^{2}{\left(n^{2}+2\right)}^{2}kT\beta_{T}$$
where *n* is the refractive index of PS, *k* is the Boltzmann constant, and $\beta_{T}$ is the isothermal compressibility at *T_g_*. The excess angle-dependent $V_{V}^{ex}(\theta )$ results from the light scattering between dipoles induced by intersegmental correlations due to the ordering of segmental units. We can obtain the theoretical $V_{V}^{ex}(\theta )$ based on the exponential correlation function introduced by Debye–Bueche [49] as:(17)$$V_{V}^{ex}(\theta )=\frac{{8\pi }^{2}\langle \eta^{2}\rangle a^{3}}{\varepsilon_{0}^{2}\lambda^{4}\left(1+\nu^{2}a^{2}s^{2}\right)}$$
where ν=2π/λ, $\varepsilon_{0}$ is the permittivity of glassy PS, $\langle \eta^{2}\rangle$ is the mean square of density fluctuations, and a is the correlation length. Debye–Bueche plots for a series of PSs with different annealing times give the correlation length a value (see Figure 10).

The a value approximately decayed from 300 nm to 100 nm. This is consistent with the results of light and X-ray scattering data by Fujiki et al. [50], where a glassy PS was shown to possess heterogeneities with 100–400 nm. Consequently, the non-thermal heterogeneous structure evolved in initial PS glass is considered to reduce toward a homogeneous structure as the annealing time increases. This is related to the structural relaxation during the annealing process. In addition, the structural relaxation is suggested to be associated not with the primary *T_g_* but with the sub-*T_g_* relaxation by comparison to Figure 7. 

## 5. Discussion

The dynamic moduli E′(ω) and E″(ω) data in the glass transition zone from each master curve can be converted to the retardation spectrum $L\left(\ln\tau \right)$ using Shwarzl–Staverman two-order approximation [51,52] given by:(18)$$L\left(1/\omega \right)=\frac{2}{\pi }{\left[D\prime \prime (\omega )-\frac{d^{2}D\prime \prime (\omega )}{d{\left(\ln\omega \right)}^{2}}\right]}_{\tau =1/\omega }$$
and
(19)$$D\prime \prime (\omega )=\frac{E\prime \prime (\omega )}{{E\prime (\omega )}^{2}+{E\prime \prime (\omega )}^{2}}$$

Substituting the experimental $L^{N}\left(\ln\tau \right)$ data into Equation (12) provides us with the theoretical normalized $C_{p}^{N}(T)$ curves based on dynamic viscoelasticity. The experimental $C_{p}^{N}(T)$ curves measured by DSC (Figure 4) are shown in Figure 11, together with the theoretical ones determined from DMA spectra. 

The peak intensity of $C_{p}^{N}(T)$ by DSC measurements increases with increasing annealing time, whereas $C_{p}^{N}(T)$ estimated from DMA is independent of the annealing time which is because the activation energy of the primary *E*″ peak (*T_g_*) is almost constant. As a result, although the $C_{p}^{N}(T)$ curve by DSC was in accordance with the $C_{p}^{N}(T)$ curve calculated from DMA spectra only for the non-annealed PS sample, the difference between experimental and calculated heights of $C_{p}^{N}(T)$ peaks was enhanced more and more with the annealing time. This suggests that the annealing effect of sub-*T_g_* on the *E″* spectra dominantly contributes to the overshoot in the specific heat in the heating process.

When the original TNM phenomenological model quantitatively fits experimental endothermic data, the non-exponentiality plays a central role in describing the experimental specific heat curves [16]. The non-exponentiality is likely related to the sub-*T_g_* process which broadens the *T_g_* retardation peak. The evidence of the existence of this distribution was demonstrated by Kovac et al. [38,53] with memory effects which showed that the response of the glass is a function of its overall previous thermal and mechanical history. The total loss of the activation enthalpy ${∆H}_{\mathrm{sub}T_{g}}\;$ of sub-*T_g_* is given by the sum of activation energy loss of both sub-*T_g_*_1_ and sub-*T_g_*_2_.
(20)$$∆H_{\mathrm{sub}T_{g}}={\left|H_{1}(t)-H_{1}(0)\right|}_{subT_{g1}}+{\left|H_{2}(t)-H_{2}(0)\right|}_{subT_{g2}}$$

The relationship between enthalpy loss $∆H_{\mathrm{DSC}}$ of DSC endotherms and ${∆H}_{\mathrm{sub}T_{g}}$ is compared in Figure 12a.

The unit of $∆H_{\mathrm{DSC}}$ was converted from J/g to J/mol using the M_n_ value. The normalized values of enthalpy loss of DSC endotherms were previously plotted against the annealing time, as shown in Figure 4. The linear relation between the enthalpy loss $∆H_{\mathrm{DSC}}$ and ${∆H}_{\mathrm{sub}T_{g}}$ with slope equal to unity passed almost through the original point, indicating that the enthalpy relaxation process was well correlated to the sub-*T_g_* relaxation process on the dynamic mechanical spectra. Furthermore, the normalized enthalpy loss $∆H^{N}$, as shown in Figure 4, was found to be proportional to the enthalpy loss $∆H_{\mathrm{DSC}}$ of DSC (see Figure 12b). The conversion factor obtained from its slope was 29.0 kJ/mol K. Consequently, the activation energy changes in dynamic mechanical spectra can be converted to the normalized enthalpy changes in DSC curves.

It follows that sub-*T_g_* endotherms are superimposed on the primary glass transition heat capacity ‘background’, observed at the same cooling and heating rates [54]. In addition, we assumed that the specific heat curve is expressed by a Gaussian distribution curve using normalized values of enthalpy loss of sub-*T_g_*. The calculated $C_{p}^{N}(T)$ curves involving the primary and sub-*T_g_* relaxation process on dynamic mechanical spectra are shown in Figure 13, together with the experimental normalized specific heat curves. It is confirmed that the curves calculated by the present model were almost close to the experimental ones for any annealed sample. This implies that the enthalpy recovery under the annealing process is dominated by the structural relaxation of the non-equilibrium aggregation state toward the equilibrium state.

So far, a correlation between enthalpy relaxation and mechanical behavior [33,54,55,56,57], such as viscoelastic, stress relaxation, creep compliance, and stress–strain properties, has been widely studied for various glassy polymers. Although the absolute values of enthalpy relaxation time, which are obtained from a KWW equation, and mechanical characteristic time differ, there exists an empirical correlation between calorimetric and viscoelastic changes during structural relaxation under annealing processes. Sasaki et al. [58] found that enthalpy relaxation and dielectric relaxation evaluated by the same stretched exponential function (KWW equation) are almost identical. However, there exists no consensus with regards to aging or annealing effects for enthalpy or density and mechanical properties despite their same molecular origin. Pye et al. [59] and Cangialosi et al. [60] demonstrated that glass transition for PS occurs on multiple time scales and occurs via a two-step process. In this work, the use of the viscoelastic retardation spectra instead of the KWW function made it possible to relate the enthalpy relaxation behavior to the viscoelastic behavior for annealed PS samples. The annealing effects for sub-*T_g_* processes dominated the enthalpy recovery process for annealed PS systems, implying that the segmental motions locally frozen in the non-equilibrium conditions are released under annealing or aging. Thus, it is likely that enthalpy recovering is dominated mainly by the secondary dispersion, i.e., sub-*T_g_*, while the overall mechanical properties are dominated mainly by the primary dispersion, i.e., *T_g_*. Consequently, the differences in $C_{p}^{N}$ between DSC and DMA, including only the primary *T_g_* relaxation, become larger as the annealing time increases, as shown in Figure 11.

According to the light scattering results (see Figure 10), the non-thermal heterogeneous structure with a size of a few 100 nm evolved in initial PS glass, which reduced toward homogeneous structure as the annealing time increased. This structural relaxation process governs the sub-*T_g_* dispersion. The fact that the relaxation of the inhomogeneous structure toward the equilibrium is caused by the annealing process lowers the relaxation strength of sub-*T_g_* with the annealing time. Hence, the reduction of the activation energy of sub-*T_g_* due to the annealing reflects the relaxation of the non-equilibrium structure, resulting in a decrease in the activation enthalpy of sub-*T_g_* corresponding to the enthalpy recovery in endotherms. The molecular understanding of the enthalpy recovery phenomena is illustrated in Figure 14.

## 6. Conclusions

In this work, the annealing time dependence of enthalpy recovery of polystyrene (PS) annealed at 80 °C was studied by the modified Tool–Narayanaswamy–Moynihan (TNM) model with creep retardation parameters determined from dynamic mechanical spectra. In this work, we accepted the viewpoint that the enthalpy recovery behavior of glassy PS has a common structural relaxation mode with linear viscoelastic behavior. As a consequence, the retardation spectrum evaluated from the dynamic mechanical spectra around the primary *T_g_* peak was used as the recovery function of the endotherms overshoot of specific heat. In addition, we identified shoulder peaks at some lower temperatures than the *T_g_* peak, and the sub-*T_g_* was suggested to be ascribed to the non-thermal perturbed structures observed from light scattering measurements. It was found that the activation enthalpies of primary *T_g_* are independent of annealing time, whereas that of sub-*T_g_* exponentially decreased with increasing annealing time. The activation process of sub-*T_g_* is dominated by the structural relaxation of the perturbed structures toward the homogeneous state reduction. Consequently, the enthalpy recovery of annealed PS can be quantitatively described by the use of kinetic data of the sub-*T_g_* relaxation process in addition to the retardation spectra of the primary *T_g_* in place of the use of some fitting parameters of the TNM model. This implies that the enthalpy recovery under annealing process is dominated by the structural relaxation of the non-equilibrium aggregation state toward the equilibrium state.

The use of the characteristic time distribution obtained via the viscoelastic retardation spectra, including sub-*T_g_*, allowed us to predict the enthalpy recovery behavior of annealed PS systems. We believe that this makes it possible to give a unified description of the thermodynamical behavior of glassy polymer solids. Moreover, the present work takes an important step towards a structural understanding of the changes in mechanical properties during physical aging encountered in practice.

## Figures and Tables

**Figure 1 polymers-15-03590-f001:**
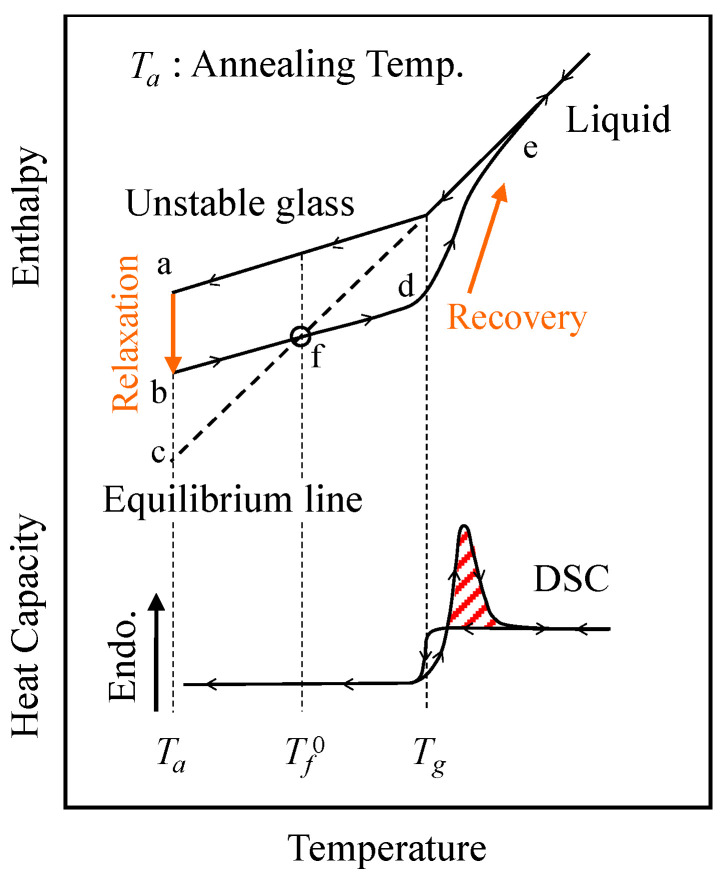
Schematic plots of enthalpy and heat capacity during cooling and subsequent heating at a fixed rate for a glassy polymer.

**Figure 2 polymers-15-03590-f002:**
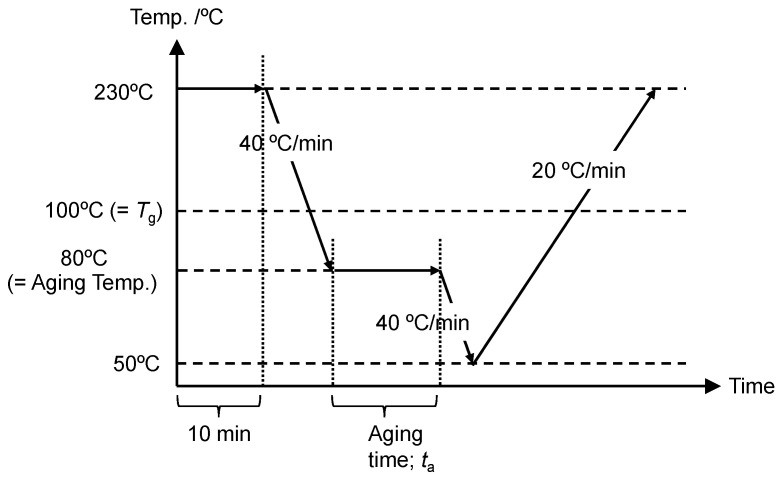
Schematic illustration of temperature scanning in aging experiment.

**Figure 3 polymers-15-03590-f003:**
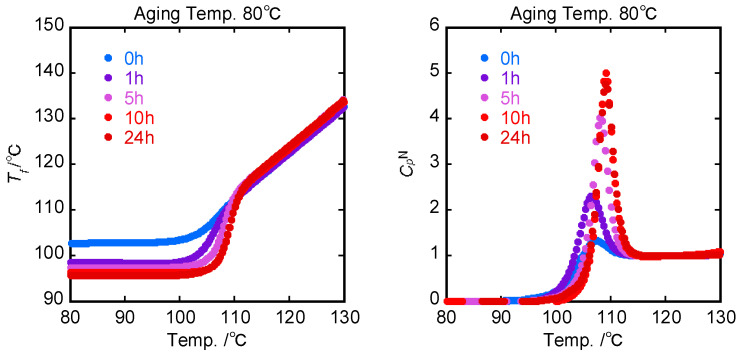
Annealing time dependence of fictive temperature and normalized heat capacity curves at an annealing temperature of 80 °C.

**Figure 4 polymers-15-03590-f004:**
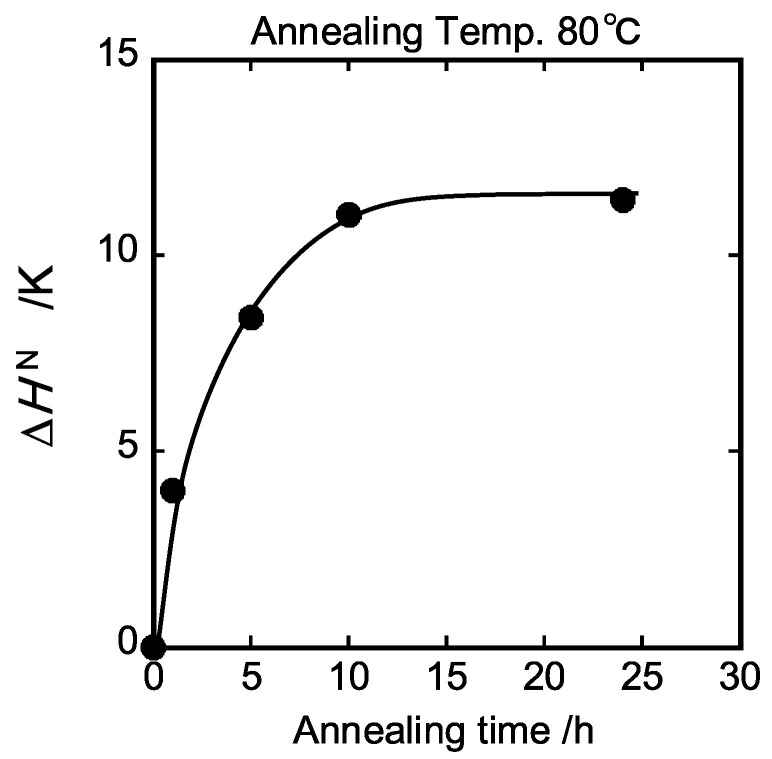
Annealing time dependence of the normalized change of enthalpy relaxation.

**Figure 5 polymers-15-03590-f005:**
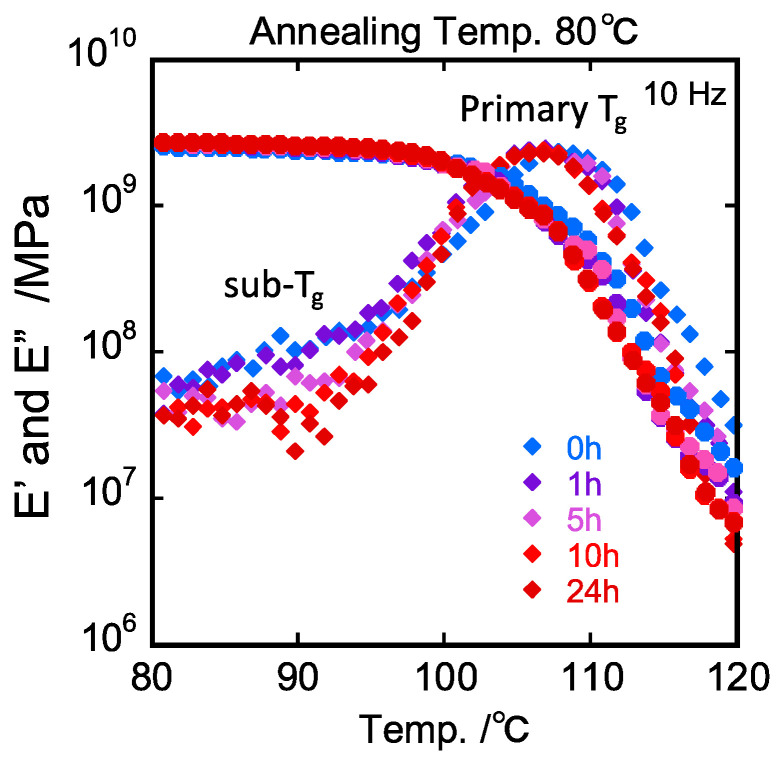
Annealing time dependence of dynamic mechanical spectra measured at 10 Hz for PS annealed at 80 °C.

**Figure 6 polymers-15-03590-f006:**
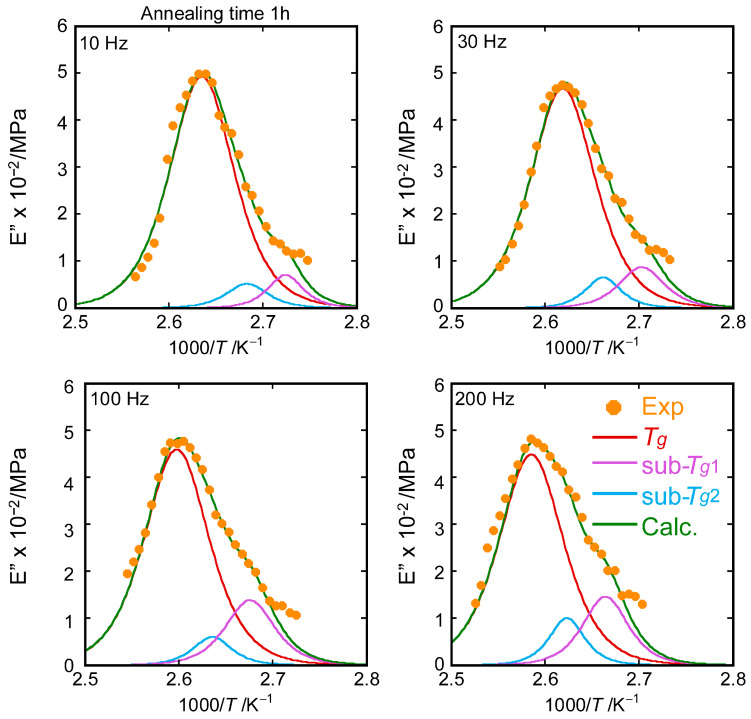
Curve fitting for dynamic mechanical spectra of loss modulus *E*″ around glass relaxation region at 1, 30, 100, and 200 Hz for PS annealed for 1 h at 80 °C.

**Figure 7 polymers-15-03590-f007:**
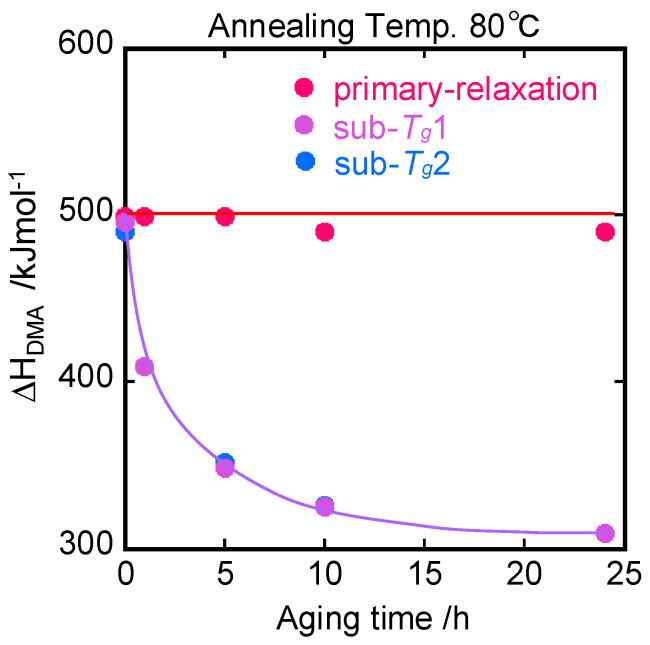
Annealing time dependence of the activation energies of *T_g_* and sub-*T_g_*.

**Figure 8 polymers-15-03590-f008:**
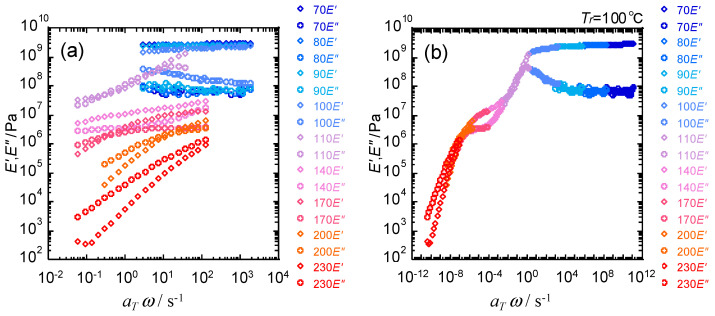
Frequency dependencies of (**a**) storage elastic modulus (*E*′) and loss elastic modulus (*E*″) and (**b**) their master curves at $T_{r}=100$ °C.

**Figure 9 polymers-15-03590-f009:**
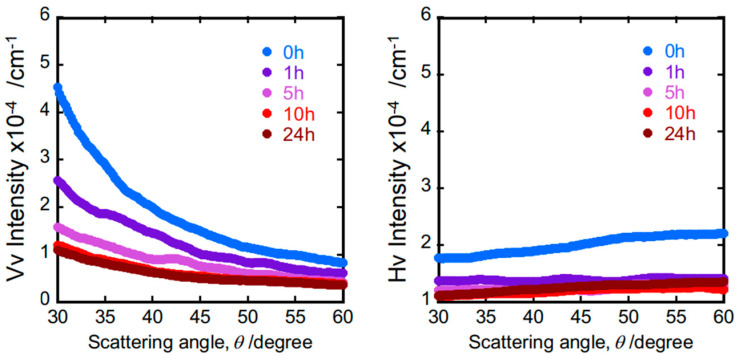
Scattering angle depends of *V_V_* and *H_V_* scattering intensities.

**Figure 10 polymers-15-03590-f010:**
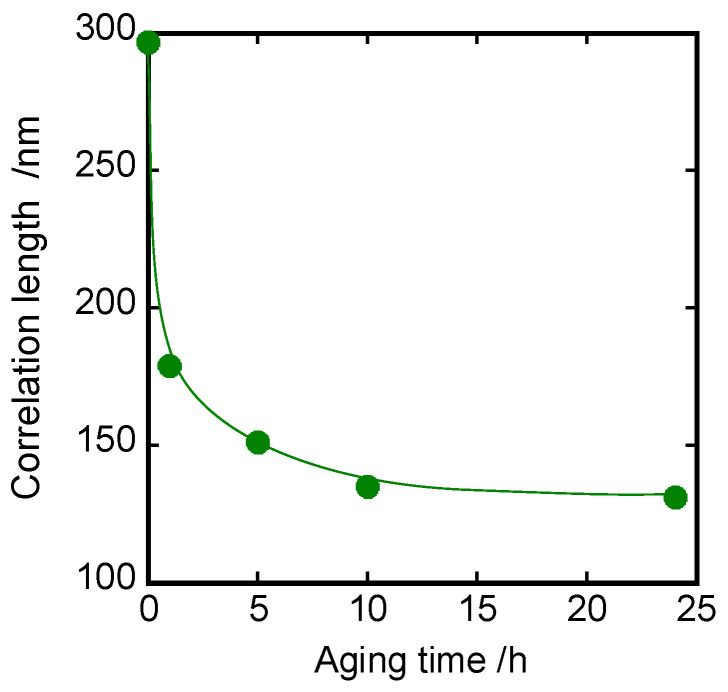
Annealing time dependence of correlation length.

**Figure 11 polymers-15-03590-f011:**
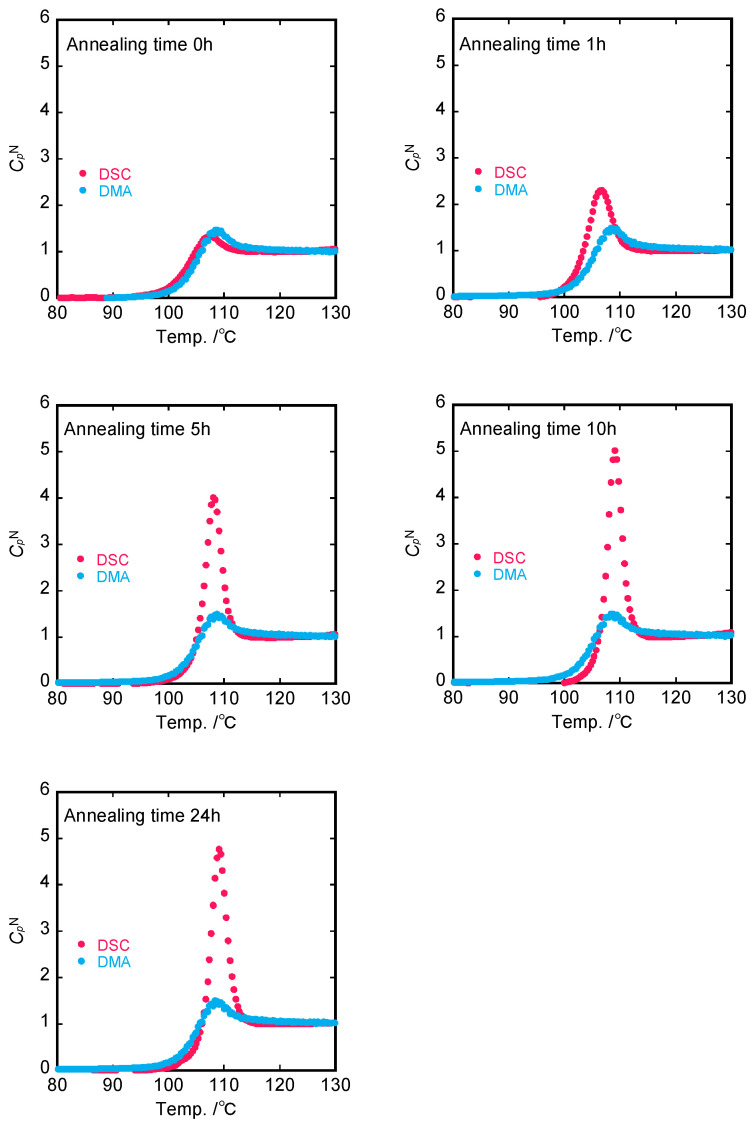
Comparison of normalized heat capacity evaluated from DSC and DMA (primary *T_g_*).

**Figure 12 polymers-15-03590-f012:**
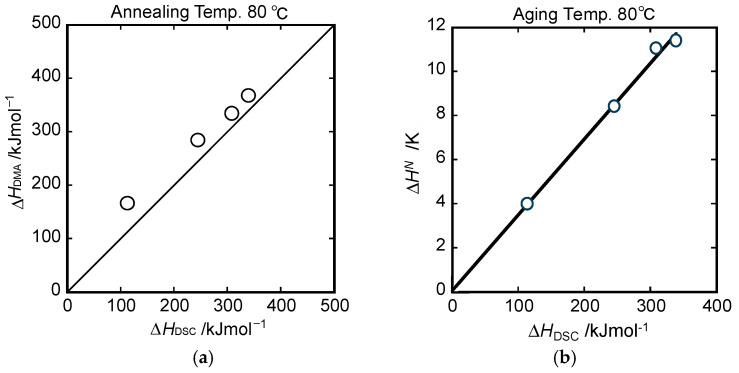
(**a**) Comparison of (**a**) enthalpy loss evaluated from DSC and activation energy of sub-*T_g_* and (**b**) enthalpy loss $∆H_{\mathrm{DSC}}$/kJmol^−1^ and normalized enthalpy loss $∆H^{N}$/K.

**Figure 13 polymers-15-03590-f013:**
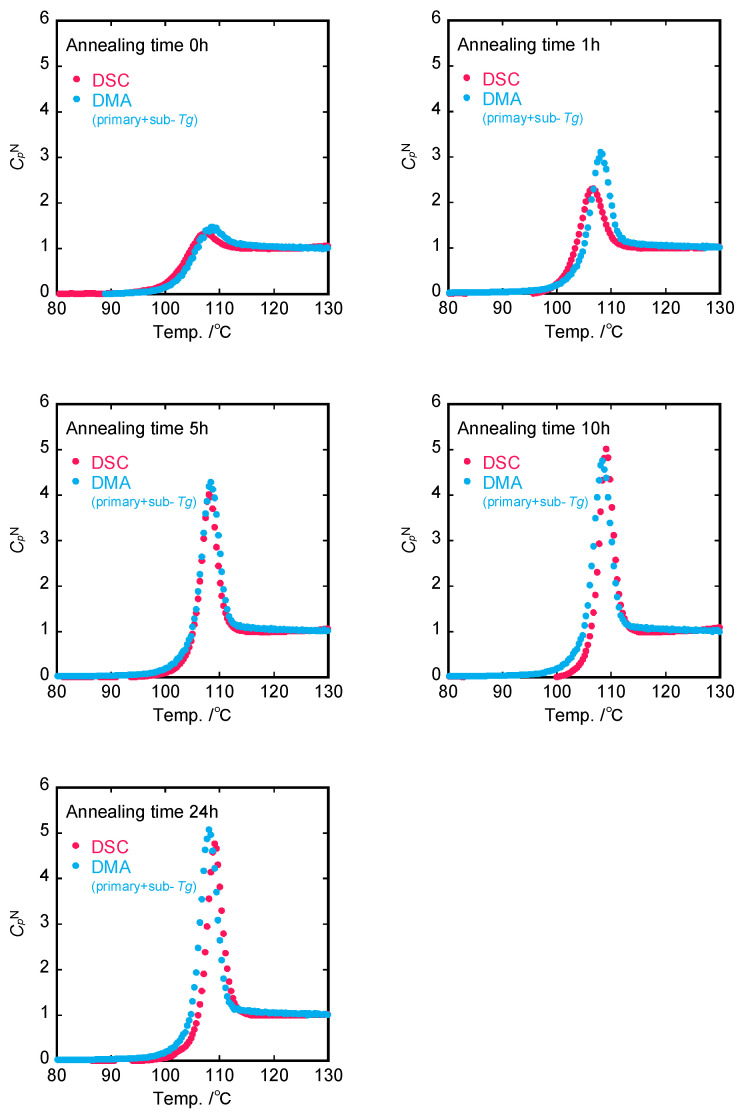
Comparison of normalized heat capacity evaluated from DSC and DMA (both sub-*T_g_* and primary *T_g_*).

**Figure 14 polymers-15-03590-f014:**
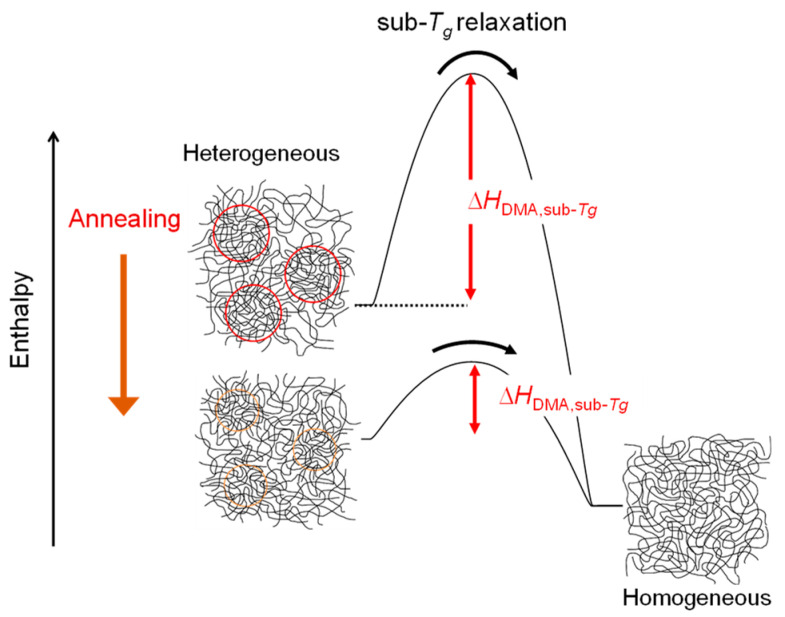
Schematic illustration of sub-*T_g_* relaxation process.

## Data Availability

The data that support the findings of the study are available on request from the corresponding author.

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
