# Peer review of "A Phenomenological Model for Enthalpy Recovery in Polystyrene Using Dynamic Mechanical Spectra"

_polymers, 2023, doi:10.3390/polym15173590_

Round 1
Reviewer 1 Report (Previous Reviewer 2)
I appreciate the effort you have put into revising the manuscript. However, there are still some crucial aspects that need further attention and clarification. I kindly request that you address the following comments to enhance the overall quality of the manuscript:
1. Based on your reference[47], equation 15 seems to be incorrect. It should be 4/3*Hv not 3/4*Hv; Please check all possible mistaken from this equation too.
2. Citation for Equation 16: Despite my previous comment, the citation for Equation 16 remains unchanged. I was unable to locate Equation 16 in the provided reference [48]. Please double-check this citation and ensure that all references throughout the manuscript are cited accurately.
3. Discrepancy in Equation 18: I must reiterate a previously unresolved comment regarding Equation 18. I have compared your Equation 18 with the related reference by Shwarzl-Staverman [51]. While your Equation 18 is similar to Equation (6c) in Shwarzl-Staverman [51], they utilize E'', whereas you introduce the newly defined parameter D''. I kindly request that you provide further clarification regarding this discrepancy, either through textual explanation or appropriate citation, to ensure a clear and consistent understanding of the mathematical framework.
Author Response
Please see the attachment.

Reviewer 2 Report (Previous Reviewer 4)
Accept in present form.
Author Response
Thank you for your review report.
Reviewer 3 Report (New Reviewer)
The manuscript aims to describe physical aging, monitored via enthalpy recovery experiments, of an archetypal glassy polymer, polystyrene (PS), substantially below Tg accounting for the typical signature of such process, that is, non-exponentiality and non-linearity. Rather than assuming a-priori a functional form to describe the aging time variation of the relaxation time and its distribution (as done for instance in the TNM model), the authors directly determine this from dynamic mechanical experiments. Specifically, from the temperature/aging time dependence of the storage and loss moduli, E’ and E’’, respectively, the spectrum of retardation time is derived, which subsequently is employed to predict Tf and, therefore, the specific heat as a function of temperature and aging time. Furthermore, the study presents a characterization of the glass heterogeneity evolution during aging by light scattering.
The present work is very interesting from both a conceptual viewpoint and the important outcome of the study. In particular, an insight of utmost importance is that considering exclusively the main relaxation does not suffice whatsoever to describe physical aging of PS at 80 ºC; and secondary relaxations must be accounted for. The latter result is in line with recent findings, inspired by the seminal work contained in ref. [59]. I warmly recommend publication of the manuscript. However, prior to that, the manuscript must be substantially improved addressing the following points:
1. The manuscript reports no information on how the experimental Tf was determined (by the Moynihan method I guess) nor how the baseline difference between aged and reference scans was accounted for. This must be reported in detail.
2. The aging time dependent activation enthalpy for different relaxation processes is presented in Figure 6. However, Figure 5 only reports dynamic-mechanical spectra after 1 hrs aging. It will be way more instructive and transparent to show these results after different aging times.
3. In Figure 3, I suggest to use log aging time scale. This would allow checking whether the glass is approaching a plateau with partial enthalpy recovery as for reported in ref. [59].
4. The treatment in equation 8 and 9 assumes that Boltmann principle applies to non-linear response, where it is actually strictly so only for linear perturbation. Please, discuss.
5. Line 294: I guess “The angle-independent” should read “The angle-dependent”.
6. Provide axes names in Figure 8.
7. I suggest reading the follow-up articles of Ref. [59], which contain further development in the same line of the present manuscript.
8. I expect further developments of the present study; for instance aging for longer times, where the main relaxation related behavior sets over (see ref. 59) or at different temperatures.
Some not very serious improvements on the English language are warranted
Round 2
Reviewer 3 Report (New Reviewer)
The revised manuscript presents significant improvements with respect to the previous version. Before publication, the description of how Tf is determined needs to be improved. The Moynihan or the matching area method, which is used to determine Tf from measurements of the specific heat, must be explained in details, also presenting the equation used to this aim.
Author Response
Response to Reviewer #3
The revised manuscript presents significant improvements with respect to the previous version. Before publication, the description of how Tf is determined needs to be improved. The Moynihan or the matching area method, which is used to determine Tf from measurements of the specific heat, must be explained in details, also presenting the equation used to this aim.
It has been identified that we have a misunderstanding of how Tf is determined. Therefore, we have newly inserted schematic plots of enthalpy and specific heat (DSC) as Figure 1 in order to clarify the essence of the aging process. On the basis of the schematic plots, we have modified some sentences in lines 230-233 in the revised MS. The actual procedure was shown as Figure S-1 in the supporting information to help the reader's understanding of the process.
ORIGINAL: The initial fictive temperature T0 was determined as the extrapolated intersection of the pre-transition (or glass region) and post-transition (or liquid region) DSC baselines in enthalpy units. The fictive temperature as shown in Figure 2 was obtained by adding T0 to the integration with time (or temperature) using Equation (7).
REVISED: The limiting fictive temperature [27] was determined as the extrapolated intersection of the pre-transition (or glass region) and post-transition (or liquid region) in enthalpy units as indicated by point (f) of Figure 1. The method for determining of PS annealed for 24 h at 80 °C was exemplified in Figure S-1. The fictive temperature as shown in Figure 3 was obtained by adding to the integration with time (or temperature) using Equation (7).

This manuscript is a resubmission of an earlier submission. The following is a list of the peer review reports and author responses from that submission.
Round 1
Reviewer 1 Report
Dear Editor,
The article by Nitta et al. concerns an investigation of the dependence of the specific heat enthalpy of glassy polystyrene (PS) on the annealing time via a phenomenological model. Authors correlate the enthalpy recovery behavior of PS with its linear viscoelastic behavior.
The topic is important and the results of the current work could be useful to other researchers in the field. I would recommend publication after a few important points are clarified.
1) As it is also mentioned in the article, the relationship between enthalpy relaxation and viscoelastic, or mechanical, response, have been extensively studied in the literature. What I found missing in the manuscript is a detailed discussion of the presented data in accordance to results from previous works in the literature.
I would also recommend Authors to provide, in the last section, a more thorough discussion of the new findings of the current work and their implications for future studies.
2) Figures 10 and 11: The comparison between the enthalpy loss of DSC and activation enthalpy change of DMA is very interesting. However, I found the discussion rather limited. In some cases there is a very good agreement between DSC and DMA values, whereas as the annealing time increases larger differences between the two quantities appear.
3) In the current study annealing times up to 24h have been used. Nevertheless, aging of polymers might be strong even after much longer times. It would be interesting, if possible, to examine systems using longer annealing times, or at least to critically discuss such potential issues.
4) It would be helpful to provide more details about the calculation of E’ and E” form the DMA experiments. Why not presenting the full E’(ω) and E”(ω) spectra?
5) Values of E”(ω) in Figures 4 and 5 do not seem compatible to each other.
Moderate editing of English language is required
Reviewer 2 Report
I recommend that the authors enhance the overall quality of the manuscript by addressing the following comments:
1. The introduction could benefit from a clearer roadmap of the paper, outlining the main sections and what readers can expect to find in each section. This could help readers to better navigate the paper and find the information they are looking for.
2. In Figure 2, the value of Tf is evaluated using Eq. 6. However, to use Eq. 6, it is necessary to know the characteristic time τ0. I suggest you explain in more detail how τ0 and Tf are determined. Additionally, I noticed that you have derived a new expression for Tf in Eq. 12. In light of this, I am curious why Eq. 6 is still being used in Figure 2.
3. It appears that the citation for Eq. 14 and Eq. 15 is incorrect. I was unable to find Eq. 14 and 15 in the given reference [47]. Please verify the correct source for these equations. In addition, if my understanding is correct, there is a typo, you should type E'' instead of E'.
4. The caption for Figure 7 is not centered. Please ensure that the caption is aligned properly with the figure.
5. It seems that the citation for Eq. 16 is also incorrect. I was unable to find Eq. 15 in the provided reference [48]. I kindly request that you double-check all citations throughout the manuscript for accuracy.
6. Please use standardized terminology for the different figures. For example, in Figure 9, the caption refers to "heat capacity," while the x and y-axes use "enthalpy loss." Similarly, in Figure 10, the caption mentions "enthalpy change," but the y-axis employs "heat capacity ." It would be helpful to use consistent terminology within each figure for clarity and coherence.
7. On line 341, you reference the wrong figure. It should be Figure 3 instead of Figure 4. Please correct this citation accordingly.
8. I have compared your Equation 18 with the related reference by Shwarzl-Staverman [47]. While your Equation 18 is similar to Equation (6c) in the paper by Shwarzl-Staverman, they utilize E'', whereas you introduce the newly defined parameter D''. Please provide further clarification regarding this discrepancy.
Reviewer 3 Report
1. Usually, physical aging occurs when the amorphous sample is kept below the glass transition temperature (15 units below Tg). Why did you store the sample above the Tg?
2. How was the temperature was annealed selected for the sample ? Why 80C?
3. On line 73, " hearing/cooling " should be changed to " heating/cooling"
Reviewer 4 Report
1) What was the nitrogen flow rate in ml/min in DSC measurements?
2) Schematic illustration of temperature scanning in aging experiment (Figure 1.) in temperature 50oC has a description “No hold” but there is a horizontal segment which suggest a short isotherm in this temperature.
3) The sample was kept at 230oC for 10 min and cooled to samples temperature 80oC (=Tg-20oC) at 40oC/min, keeping for annealing time of 1, 5, 10, and 24 hrs, further cooled to 50oC and then heated to 230oC at 20oC/min for monitoring enthalpy recovery.
a) Why the measurement were not started from annealing temperature (80oC)?
b) Why the samples were cooled to 50oC???
c) Why the samples were annealed only in one temperature (80oC)?
4) I believe that all errors will be corrected:
a) The DSC experiment under continuous hearing… (line 73)
b) The DSC measurements were performed using a Parkin-Elmer Diamond-DSC.. (line 165)
c) sealed in the aluminium pans. (line 169)
d) Figure 2. Aging time dependence of Fictive temperature and normalized hear capacity curves (line 216)
e) annealed at 80C (line 372)